# Genistein: A Potent Anti-Breast Cancer Agent

Smitha S. Bhat [1,†] , Shashanka K. Prasad [1,†] , Chandan Shivamallu [1] , Kollur Shiva Prasad [2] , Asad Syed [3],
Pruthvish Reddy [4] , Charley A. Cull [5] and Raghavendra G. Amachawadi [6,*]

1 Department of Biotechnology and Bioinformatics, Faculty of Life Sciences, JSS Academy of Higher Education and Research, Mysuru 570015, Karnataka, India; smithasbhat@gmail.com (S.S.B.); shashankaprasad@jssuni.edu.in (S.K.P.); chandans@jssuni.edu.in (C.S.)
2 Department of Sciences, Amrita School of Arts and Sciences, Amrita Vishwa Vidyapeetham, Mysuru Campus, Mysuru 570026, Karnataka, India; shivachemist@gmail.com
3 Department of Botany and Microbiology, College of Science, King Saud University, P.O. Box 2455, Riyadh 11451, Saudi Arabia; assyed@ksu.edu.sa
4 Department of Biotechnology, Acharya Institute of Technology, Bengaluru 560107, Karnataka, India; pruthvi.19@gmail.com
5 Midwest Veterinary Services, Inc., Oakland, NE 68045, USA; charley@mvsinc.net
6 Department of Clinical Sciences, College of Veterinary Medicine, Kansas State University, Manhattan, KS 66506, USA
* Correspondence: agraghav@vet.ksu.edu
† Equal first authors: Smitha S. Bhat and Shashanka K. Prasad.

**Abstract:** Genistein is an isoflavonoid present in high quantities in soybeans. Possessing a wide range of bioactives, it is being studied extensively for its tumoricidal effects. Investigations into mechanisms of the anti-cancer activity have revealed many pathways including induction of cell proliferation, suppression of tyrosine kinases, regulation of Hedgehog-Gli1 signaling, modulation of epigenetic activities, seizing of cell cycle and Akt and MEK signaling pathways, among others via which the cancer cell proliferation can be controlled. Notwithstanding, the observed activities have been time- and dose-dependent. In addition, genistein has also shown varying results in women depending on the physiological parameters, such as the early or post-menopausal states.

**Keywords:** genistein; apoptosis; breast cancer; cell cycle; estrogen receptor

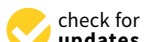



## 1. Introduction

Genistein, an isoflavone, is a natural phytoestrogen present in soybeans and native to Southeast Asia. It was first isolated from *Genista tinctoria (L.)* in 1899 and named after it, following which it has been mostly identified in the *Trifolium* spp., exclusive to the Leguminosae (Fabaceae) [1].

Several in vitro and in vivo studies have attempted to understand and gain a better insight into the mechanisms underlying the biomedical properties of genistein [2–4]. The isoflavonoid has been analyzed and previously reviewed for its neoplastic potentials. The pathways though which genistein alleviates breast cancer include various grey areas which pertain to the molecular mechanisms of genistein, and preclinical results remain unclear. The identification of the mechanistic action of genistein on breast cancer could help in the development of anti-breast cancer therapy in cases where there are no targeted therapies known or available. Further research into the mechanistic action of genistein could lead to the development of a potential plant-based cancer drug with minimal deleterious effects, along with overcoming drug resistance and repression of reoccurrence of cancers. Such a development of genistein in chemotherapy may be a powerful tool in personalized medicine. The current review focuses mainly on the molecular basis of the anti-breast cancer potential of genistein, wherein we have complied the evaluations of the pathways and various targets of this molecule when administered.

## 2. Chemistry of Genistein

### 2.1. Structure

In plants, the synthesis of genistein starts from a flavanone, naringenin, by the isoflavone synthase enzyme due to ring migration [1,3]. The structure of genistein (chemically, 4′,5,7-trihydroxyisoflavone (C15H10O5)) and estradiol have been observed to be similar [5]; hence, genistein has estrogenic activity and is a good example of a phyto-estrogenic substance. Its nucleus is made up of two arenes (A and B) coupled to another carbon ring (C). It has a limited water solubility and a preference for polar solvents such as acetone and ethanol. It has a C2-C3 double bond in its basic carbon skeleton, as well as an oxo-group in the C ring at the C4 position along with 3 hydroxyl groups at the C 4′, 5, and 7 locations of rings A and B [4]. The structure of genistein is illustrated in Figure 1.

**Figure 1.** Structure of genistein. PubChem CID 5280961 (https://pubchem.ncbi.nlm.nih.gov/compound/Genistein, accessed on 1 October 2021).

### 2.2. Synthesis of Genistein

Baker was the first to synthesize genistein organically in 1928 [6] using deoxybenzoin as a substrate. The cyclization of ketones was used as a chemical method of genistein synthesis in an oven [7]. Its synthesis from 2,4,6-trihydroxyphenyl ethenone with the two hydroxyl substituents in the triol as methoxymethyl ester has been attempted using a technique that begins with ketone production, followed by closing of the ring structure and a Suzuki coupling reaction with palladium acetate and polyethylene glycol [8]. Treatment of trihydroxybenzoin, derived by acylation of phloroglucinol substituted with phenyl acetonitrile using hydrochloric acid and zinc chloride with catalyst dry ether, is a more contemporary technique to genistein production [9]. Biotechnological synthesis was accomplished by converting (2S)-naringen to genistein under NAD(P)H and oxygen-dependent states and adding cytochrome P-450 to soybean cell cultures [10]. Employing genetically modified *Saccharomyces cerevisiae* cells containing the isoflavone synthase gene obtained from *Glycyrrhyza echinata*, a metabolic approach along with engineering tools was set up as genistein synthesis [11]. In *Nicotiana tabacum* leaves transformed with IFS, genistein was created via acting on the phenylpropanoid pathway; however, ultraviolet ray treatment also increased genistein assembly [12]. Biological genistein synthesis from p-coumaric acid or naringenin was attempted utilizing *Escherichia coli* as a biotransformation host using Os4CL, PeCHS, RcIFS, and OsCPR for production [13].

### 2.3. Synthesis of Genistein Derivates or Analogues

Synthesis of analogues of genistein was achieved by the Ferrier rearrangement of compounds yielding 2,3-unsaturated bromo-alkyl-glycosides, which were then epoxidated with meta-chloroperoxybenzoic acid before coupling with genistein [14]. For the manufacture of genistein derivatives, new glycosylation and glycoconjugation chemical techniques have been devised [15]. A novel three-step synthesis from genistein of a water-soluble compound was also attempted, in which base-catalyzed reaction of genistein was hydrolyzed to obtain the target compound [16].

### 2.4. Bioavailability and Metabolism of Genistein

The amount of a component that is absorbed in the body is known as bioavailability. It is critical to research a chemical's bioavailability in order to determine how effective it is on the body. Poor water solubility of genistein is a limitation to overcome for its bioavailability after oral administration, for which water-soluble derivatives of genistein were synthesized [17]. Because of its low molecular weight (270 kDa) and lipophilic characteristics, genistein is quickly absorbed in the intestine in both rodents and humans [18]. A very low half-life of approximately 46 h was observed in vivo following oral administration [19]. Glucuronidation and sulfation are major pathways of metabolism of genistein with the production of metabolites [18]. Once consumed, genistein is converted into genistein glucuronide and sulphate in the intestine, which along with genistein circulate through veins with the assistance of multidrug resistance-associated protein 3 transporters with a 100% absorption ratio [20]. The metabolites are excreted through bile or through kidneys. In humans, micromolar levels of genistein in blood can be found through prolonged dietary exposure [20,21]. Metabolomic studies may be required in order to assess the intracellular concentrations of genistein at which modulation of a range of targets occur and hence, careful attention is required towards the dose-dependent behavior of genistein, as well as the pertaining molecular intricacy [22,23]. One main limitation with genistein being a natural compound is its low water solubility, which may need to be modified with respect to its chemical structure in order to increase solubility and have higher bioavailability [24]. Furthermore, studies may need to be performed on identifying the purified individual versus mixture of isoflavones present in breast cancer. However, studies observing the pharmacological and biomedical activity of unbound genistein in comparison with its metabolic products are less. Hence, it is important to evaluate free, unbound genistein concentration in blood. Being bitter in taste, genistein requires different formulations in order to overcome the taste, as well as the limitation of bioavailability.

## 3. Genistein and Cancer

Genistein has demonstrated a plethora of biomedical effects, such as anti-oxidation, anti-proliferation, and tumoricidal activities [25]. More importantly, in vivo, in vitro, as well as in silico research into its anti-cancer properties have pointed towards a pivotal role played by genistein as an anti-tumoricidal molecule in varied types of cancer [26]. Two very important reasons for the extensive research conducted on genistein over the past decade are the evidence of lower risk of diseases in association with its administration and to look for pharmacologic drugs that affect with growth factor signaling pathways in cells.

Numerous previous studies have reported arrest of cell-division cycle and apoptosis in multiple cancer cell lines in in vitro studies, as well as demonstration of the same in vivo [4,25]. When researchers looked at the consequences of genistein on cell cycle progression in prostate cancer cell lines, they discovered that it stopped cell-division cycles in the G2/M phases due to the downregulation of cyclin B expression, leading to the conclusion that it could be a potent regulator of cyclin B with potential applications in cancer prevention [27]. In a study of the pleiotropic molecular effects of genistein on head cancer cells, researchers discovered that genistein causes molecular alterations in the cancer cells that impede cell development and induce apoptosis. In a series of tests, the same researchers discovered that genistein halted progression through the cell cycle and death in a head cancer cell line through regulating p21WAF1 and Bax, as well as repressing cyclin B1 and Bcl-2. They further confirmed that genistein reduces metaphase chromosomal spread and hampers nuclear translocation of human telomerase reverse transcriptase without impacting telomerase activity via downregulating cerbB-2 [28]. Some recently discovered mechanisms employed by genistein in various cancer models to bring about anti-cancer effect are summarized in Table 1.

**Table 1.** Some recently discovered anti-cancer mechanisms of genistein.

| Effect | Mechanism | Cancer Model | Reference |
|---|---|---|---|
| Evasion of Apoptosis | ER-stress | HL-60 | [29] |
| | ↑ROS | Mia-PaCa2 and PANC-1 | [30] |
| Cell cycle arrest | G0/G1arrest | Mia-PaCa2 and PANC-1 | [30] |
| | Mitotic arrest, ↓PlK1 | TP53-mutated A460 cancer cells | [31] |
| Anti-metastatic | ↓DMBA-induced metastatic transition | Mouse model | [32] |
| Anti-proliferative | ↑p-ERK ↑BDNF ↓AChE | Mouse model | [33] |
| | ↓mTOR ↓p70S6K1 ↓4E-BP1 ↓Bcl-2 ↑Nrf2 ↑HO-1 ↑Bax | Hen model | [34] |
| | ↓HDACs | HeLa cells | [35] |

ER—Estrogen Receptor; ROS—Reactive Oxygen Species; PlK1—Polo-Like Kinase 1; DMBA—7,12-Dimethylbenz[a]anthracene; p-ERK—Phosphorylated Extracellular Signal-Regulated Kinase; BDNF—Brain-Derived Neurotrophic Factor; AChE—Acetylcholinesterase; mTOR—Mammalian target of rapamycin; p70S6K1—Ribosomal protein S6 kinase β 1; 4E-BP1—Eukaryotic translation initiation factor 4E-binding protein 1; Bcl-2—*BCL2* apoptosis regulator gene; Nrf2—Nuclear factor erythroid 2-related factor 2; HO-1—Heme Oxygenase 1; Bax—BCL2 Associated X, Apoptosis Regulator gene; HDACs—Histone Deacetylases.

## 4. Genistein and Breast Cancer

### 4.1. Epidemiology

Breast cancer has been classified as one of the prevailing malignancies in women throughout the globe, with the American Cancer Society estimating that over 43,600 women will die from breast cancer in 2021 [36]. Various natural compounds with pharmacological capabilities are being explored as an alternative to manufactured anti-cancer medications in order to overcome their negative side ramifications. Genistein is one such chemical. In various studies, epidemiologic data has suggested that soy consumption is oppositely proportional to the risk of breast cancer, with Asian women and men who consumed a soy diet having a 40% lower prevalence of mammary cancer, while Asians who did not consume a traditional soy-rich diet lost this protection [37,38]. However, the soy isoflavone in several in vitro and in vivo models with bone micro-metastasis in mice have been observed to stimulate breast cancer and further research in human subjects maybe required about the duration of consumption of the same by breast cancer survivors [39].

### 4.2. Mechanism

The tumoricidal effects of genistein have been seen on cell lines and in breast cancer-induced animal models at various dosages. Genistein has been linked to distinct pathways and targets. Apoptosis, cell-division cycle modification, and anti-cell proliferation are some of the strategies that have been proposed as genistein targets and pathways for anti-breast cancer tumorigenesis and are discussed below in Table 2.

**Table 2.** Some possible anti-breast cancer molecular mechanisms for genistein and its targets.

| Effect | Proteins/Pathways Affected | Reference |
|---|---|---|
| Decreased response to growth factors | Downregulation of tyrosine kinase activity | [40] |
| | Expression of SRF mRNA | [41] |
| Arrest of cell cycle | G0/G1 arrest by cell cycle transition | [42] |
| | G2/M phase arrest via cyclin B | [27] |
| Induction of apoptosis | Downregulation of CIP2A mRNA; modulation of E2F1 | [43] |
| | Activation of PPPA | [44] |
| | Inactivation of NF-kB | [44] |
| | Bcl-2 Bax | [44] |
| | Activation of Caspase-3 | [45] |
| | Upregulation of DNA fragmentation | |
| Anti-proliferative effects | Downregulation of DNA methylation Upregulation of ATM Upregulation of APC Upregulation of SERPINB5 | [46] [47] |
| | Upregulation of ER α | [48] |
| | Decreased ER binding | [2] |
| | Erβ inhibited E2-dependent cell growth | [44] |
| Cancer-associated microRNAs (mi) | miR-155—Downregulation of PTEN, casein kinase, p27 | [49] |
| | miR-23b—Upregulation of PAK2 | [50] |
| Epigenetic modifications | Tumor suppressors $p^{21}$ and $p^{16}$ c-MYC-BMI complexes Regulation of E2-induced genes | [44] |

SRF—Serum Response Factor; CIP2A—cancerous inhibitor of PP2A; E2F1—Transcription factor E2F1; PPPA—PP2C-family protein phosphatase; NF-kB—nuclear factor kappa-light-chain-enhancer of activated B cells; Bcl-2 Bax- BCL2-associated X protein; ATM—ataxia telangiectasia mutated; APC—Adenomatous Polyposis Coli; SERPINB5—Serpin Family B Member 5; ER—Estrogen Receptor; PTEN—Phosphatase and Tensin Homolog; PAK2- Serine/threonine-protein kinase PAK 2; c-MYC-BMI—myc and bmi-1 oncogenes; E2- 17β-estradiol.

### 4.3. Induction of Apoptosis

The rate of cell division rises as tumors develop, resulting in a lower rate of programmed cell death. Apoptosis can be triggered in a variety of ways, according to new research. In numerous cell lines of mammary cancer, genistein triggered apoptosis. The stimulation by the peroxisome proliferator-activated receptor gamma (PPARγ) pathway has been proposed as a possible mechanistic pathway in the prevention of mammary cancer. PPAR, PTEN, and cyclin B1 are all part of this pathway. Upregulation of PPAR expression as well as a reduction of cyclooxygenase-2 and prostaglandin E2 expression were observed when MDA-MB-231 cells were given genistein in combination with arachidonic acid, docosahexaenoic acid, and eicosapentaenoic acid, which reverted invasiveness in the cancer cells [51]. Apoptosis was observed as a result of synergistic activity of genistein combined with anti-breast cancer drugs in MDA-MB-231 cells and BT-474 cells [52,53], reducing their chemoresistance.

Apoptosis could also be instigated by calpain and caspase, which are enabled by calcium ions and mediate cell death. Depletion of calcium storage in the endoplasmic reticulum, higher $Ca^{2+}$ concentrations, activation of calpain, and hampering of calpain's $Ca^{2+}$ binding sites result in improved cytosolic $Ca^{2+}$ buffering capacity, as well as caspase inhibition, which result in a decrease of apoptosis in cancer cells. Hence, one pathway of apoptosis by genistein is through its cellular $Ca^{2+}$ regulatory activity [54]. In in vivo and in vitro models of MDA-MB-435 and Hs578t cells, as well as immunocompromised animals, mammary tumor growth was produced by hindering cell viability and eventually death of the cell [54].

When MCF-7-C3 and T47D breast cancer cells were medicated with genistein, the cancerous inhibitor of protein phosphatase 2A (CIP2A), a human oncoprotein, was dysregulated, leading to the hypothesis that CIP2A was a genistein target [43] in causing growth inhibition and apoptosis. Injection of genistein into 35-day-old rats reduced tumor size by 27%, and comparable findings were shown in nude mice bearing MCF-7 and MDA-MB-231 heterografts with mammary cancer cell invasion and tumor formation [55].

Thus, genistein has been extensively documented to induce cancer cell apoptosis via a number of mechanisms including cell-signaling pathways. Both in vitro and in vivo evidence of the apoptotic nature of genistein on breast cancer cells is highlighted, showing genistein in a promising role. However, further research may be required to recognize the intracellular targets of genistein in order to be used as a therapeutic drug.

### 4.4. Cell Cycle Arrest and Anti-Proliferative Mechanism

Nearly three decades ago, the first report revealing the hinderance of protein kinase brought about by genistein was published. Using an omics approach, genistein was discovered to regulate 183 proteins [40]. The cell-division cycle is a set of events that occur inside a cell that leads to cell multiplication and duplication. On a molecular level, genistein hinders the growth of malignant cells by acting on multiple cell-division cycle regulators and proteins. Genistein impacts cell development and progression by altering cell-division cycle-regulator proteins, such as Akt and nuclear factor [56,57]. Some proteins operate as cell division checkpoints and monitor the stages of the cell-division cycle. A balance between the regulatory proteins is required for the progression of a cell-division cycle.

One of the anti-proliferative mechanisms demonstrated by genistein is the blocking of NF-kB pathways and subsequent activation of NF-kB [57]. The EGFR/Akt/NFκB pathway modulation play a role in cell differentiation [58], which leads to cancer cell death. With genistein, the activity of Akt is suppressed, promoting the deactivation of downstream signaling pathways, including NF-κB [2,59]. This was demonstrated by the electrophoretic mobility shift assay in MDA-MB-231 cells, along with inhibition in the activation of Akt by preventing EGF signal triggering [59]. Furthermore, through modulating AMPK and COX-2, the combination of genistein and capsaicin instigated synergistic apoptotic consequences [60]. As a result, it has been concluded that genistein hinders the activation of NF-B, mostly through the inactivation of EGF and Akt or by directly deactivating it. The merging of genistein, cisplatin, docetaxel, and doxorubicin has also been shown to cause NF-kB deactivation, resulting in enhanced growth inhibition and finally apoptosis in MDA-MB-231 cells [61]. This is said to be brought about by the MEK5/ERK5 pathway [62], revoking the EGF and Akt induced NF-kappa B activation, which led to the conclusion that the inactivation of NF-kappa B cancer cells is partly arbitrated though the Akt pathway [59]. In silico studies have studied the binding interactions of active sites of these molecules, which confirmed these findings along with revelation that the amino acid residues of lysine, serine, and aspartic acid play a major role [63]. Inactivation of the Akt pathway can potentially be used to prevent proliferation [64].

In MCF-7 and MCF-7 HER2 cells, an increase in sub $G(0)/G(1)$ apoptotic fractions was observed, which could be due to induction of the extrinsic programmed cell death pathway, up-regulation of p53, reduced phosphorylation of IB, and evasion of the nuclear translocation of p65 and its phosphorylation within the nucleus [65]. MDA-MB-231 cell growth inhibition was seen in a dose-dependent manner via hindering NF-B activity via the Notch-1 signaling pathway, as well as lower production of cyclin B1, Bcl-2, and Bcl-xL [66]. Some of these mechanisms are picturized in Figure 2.

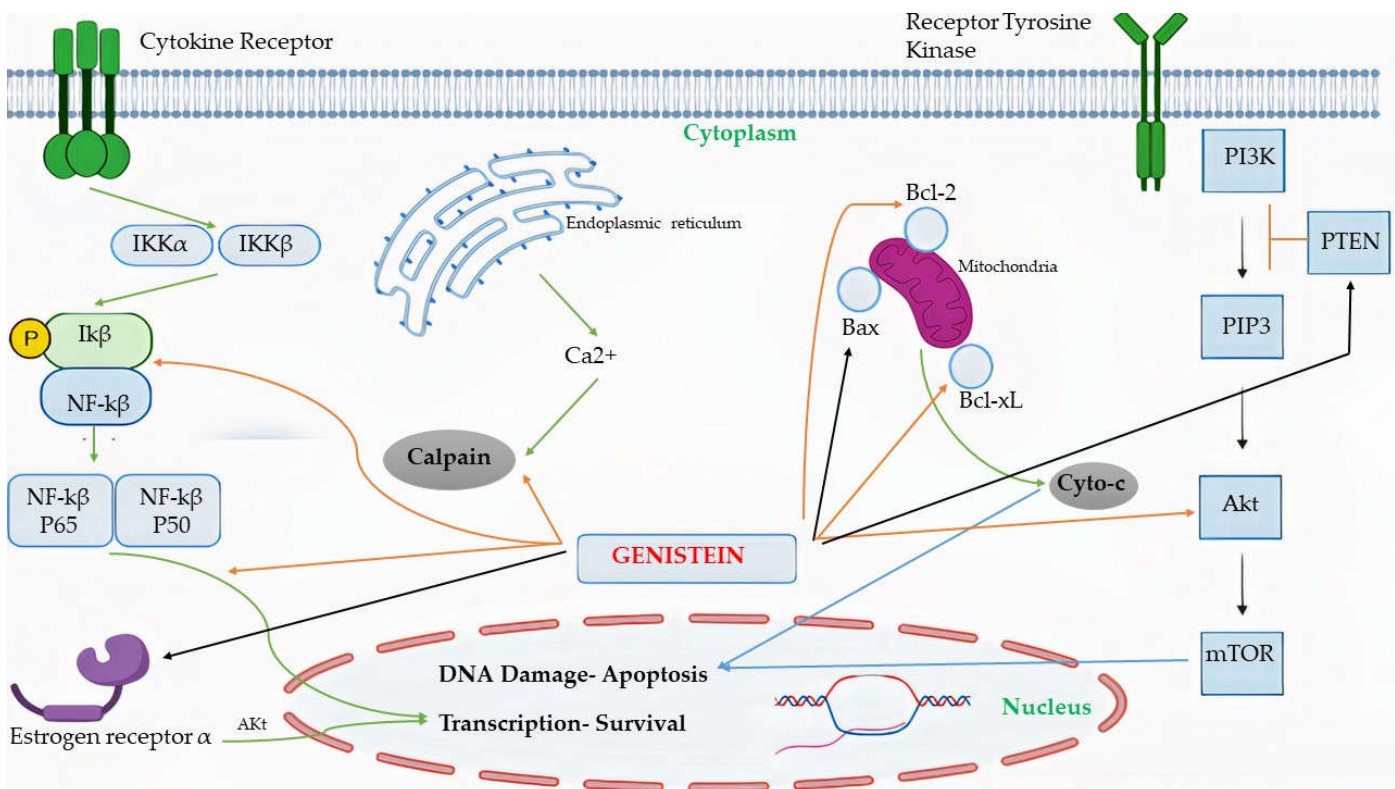

**Figure 2.** Some pathways are targets of genistein through which it affects cell survival and brings about apoptosis. PTEN—Phosphatase and tensin homolog; PI3K—Phosphoinositide 3-kinases; PIP3—Phosphatidylinositol (3,4,5)-trisphosphate; Akt—Protein kinase B; mTOR—The mammalian target of rapamycin.

Genistein causes a halt in the cell-division cycle at the G2/M phase via the expression of p21Waf11/Cip1 which is stated to have increased, eventually leading to the seize [67]. Cell-division cycle-associated phosphatase Cdc25C downregulation was also associated with genistein in MCF-10F cells [68]. Furthermore, mitogen-activated protein kinase -mediated genistein and subsequent repression of cyclin B1 and Cdc25C, as well as elevation of c-Jun and c-Fos levels, are linked to cell division arrest at G2/M phase [69]. By modulation of the RAS/RAF signaling pathway, the activation and phosphorylation of MAPK is stabilized [69]. Genistein's intrinsic stimulation of cell death is a slow process. The breakdown of the mitochondrial membrane and the generation of reactive oxygen species are caused by changes in Bcl2/Bax levels. The fundamental issue, however, is the difficulty in identifying the initial genistein target among these protein kinases.

### 4.5. Preventing Angiogenesis

Downregulation of matrix metalloprotein genes together with a decrease in cancer cell invasiveness suggests that both transcriptional modulation of genes involved in the cancer pathogenic process and repression of breast cancer cell invasiveness are linked [70]. The expression of MMPs 2, 3, 3, and 15 have been noted to be decreased in T47D cells with genistein treatment, preventing angiogenesis and metastasis [71].

Some studies also indicate that genistein is responsible for the downregulation of hypoxia-inducible factor 1-$\alpha$, with in silico backing in studies that characterized the sites of interaction between them, showing that genistein is bound to FIH-1 binding site [72].

Furthermore, in silico studies have proven the involvement of Akt, Hif1$\alpha$, and VEGF cascades in the prevention of angiogenesis by genistein [73]. The same researchers have also reported the development of spermine tethered lipo-polymeric hybrid nano-constructs in synergistic delivery of anti-breast cancer drugs and genistein by inhibition of breast

arterial calcifications. These findings could lead to possibility of finding new combinations of chemotherapeutic drugs, along with anti-angiogenic genistein using nanoparticles [73].

### 4.6. Effect of Genistein on Cancer Stem Cells

Modification in mammosphere-formation capability in breast cancer stem cells was found to be a tumoricidal targeting mechanism of genistein [74,75]. Amphiregulin released from ER+ cells activate the PI3K/Akt and MEK/ERK signaling pathways, which are connected to the mammosphere differentiation induction [74]. With the upregulation of PTEN, the signaling pathways may be inhibited, which may be a relevant pathway through which stem cells or progenitor cells may be controlled and breast cancer can be repressed [75]. Dietary exposure to genistein was found to be associated with reduced body weight, as well as adiposity in rodent models due to increased mammary tumor suppressors PTEN and E-cadherin expression [76]. Adipocyte differentiation was found to be mediated by Erβ signaling via a linear pathway that involves the activation of the Erβ and PPARγ expression [76]. Furthermore, the Hedgehog-Gli1 signaling pathway, which when blocked, lowers stem cell survival by reducing the proteins SMO and/or Gli1, has been found to be dysregulated in breast cancer stem cells [77].

### 4.7. Gene Regulation

Another mechanism by which genistein impacts breast cancer is through gene regulation. Genes involved in cell salvage were found to be increased, while genes involved with signaling pathways, cell proliferation, and differentiation were shown to be downregulated [41]. Stress response, transcription, and salvage pathway enzyme genes were all upregulated, implying that genistein is implicated in the activation of the salvage response. Genistein's anti-proliferative properties could be attributed to the stress response pathway [78]. Heat shock proteins, also known as molecular chaperones, are thought to be important for cells' adaptability to environmental changes. The induction of HSP as a result of a stress response may govern apoptotic control. Dysregulated genes include the Serum response factor (SRF), Disabled homolog 2 (DOC 2) and Fms-related tyrosine kinase 1 (flt-1) [41]. Genistein dysregulated the SRF protein, a transcription factor, and mRNA expression in a dose-dependent manner [41]. It has been proposed that genistein's inhibitory activity is due to the suppression of ER- and insulin-like growth factor-arbitrated pathways in MCF-7 cells via dysregulation of SRF expression, as SRF regulates growth factors and estrogen's non-genomic activities [79,80].

Furthermore, downregulation of genes associated with the replication of DNA such as the replication factor C 4, VJ reintegration of immunoglobulin, and T-cell genes such as the recombination activating gene 1, apoptosis, and mitochondrial synthesis of DNA occurs with treatment of genistein [41]. The downregulation of RFC4 and subsequent replication of DNA led to the identification of mechanism for the reduction in S-phase of cell-division [41]. However, the main role of all the dysregulated genes in mediating the inhibitory action of genistein remains to be determined.

The development and progression of cancer is greatly affected by cytochrome P450 1B1 (CYP1B1) via activation of estrogens and carcinogens [81]. Genistein was found to induce the CYP1B1 gene expression and hence stimulate ROS production and breast cancer cell proliferation [82]. However, more detailed studies are required in order to further assess the role played by genistein, as well as cytochrome P450 in breast cancers.

Genistein is thought to regulate epigenetic processes and thus influence gene transcription. Due to aryl hydrocarbon receptor antagonism, administration of genistein into adult female rats during conception resulted in reduced methylation of CpG in the *BRCA1* gene, as evidenced by a reduction in Cyp1b1 expression, a possible aryl hydrocarbon receptor target. Cell culture research on triple negative breast cancer cells with overexpression of active aryl hydrocarbon receptor backed up this finding. Genistein has been shown to subdue *BRCA1* expression by demethylating the *BRCA1* promoter [18,47,83]. All this data has been consistent with the other epidemiological reports available regarding the

consumption of soy products and incidence of breast cancer [37]. Genistein treatment to *BRCA1* silenced breast cancer cells, led to downregulation of GPR30 expression and the inhibition of Akt phosphorylation which induced downregulation of B1 expression, leading to cell-cycle arrest. Furthermore, the treatment also led to diminished ROS levels via upregulation of Nrf2 expression [84].

In silico studies explained that the negative effect of genistein on DNA methyltransferase may be due to competitive binding of genistein with hemi-methylated DNA at the catalytic sites of DNA (cytosine-5)-methyltransferase 1 [46,47]. Genistein has also been shown to activate the Wnt signaling pathway. In breast cancer cells, genistein treatment increased phosphorylation of βcatenin, causing it to be restricted to the cytoplasm along with downregulation of Wnt signaling and related genes such as cyclinD1 and cMyc [85]. This was proven in in vivo and in vitro studies which concluded that genistein was responsible for the inhibition activity of DNA methyltransferase (DNMT) [18], downregulation of DNA methylation, and DNA (cytosine-5)-methyltransferase 1 by its ability to demethylate and reactivate methylation-silenced tumor repressor genes [46].

Another avenue of genistein's anti-breast cancer function could be the downregulation of the estrogen receptor and its associated vascular endothelial growth factor (VEGFR). Genistein inhibits estrogen receptor expression and the processes that leads to it. VEGFR-2 expression is lowered when the estrogen receptor is inhibited [41]. Furthermore, along with enterolactone, genistein was also found to inhibit estradiol-mediated expression of VEGFR-2 [86]. Both the csf1 and VEGFR-dependent pathways have been implicated via the downregulation of DOC2 [41]. As a result, angiogenesis-related genes could be genistein's target. In an estrogen-rich environment, breast cancer cells from young or early postmenopausal women were discovered to use genistein as a replacement to grow and survive [87]. However, when breast cancer cells grew in estrogen-negative environment, i.e., in postmenopausal women, a high level of genistein was found to instigate apoptotic cell death [87]. In a 2014 clinical trial, 140 women with breast cancer at the early stages were haphazardly assigned to one of two groups and given genistein or placebo for a month. There was an over-expression of tyrosine kinase, the EGFR2 receptor, and other genes that control the cell cycle [88]. The dose-dependent nature of genistein, the time period of study, and the age range of the included women in the studies are all important factors to consider when designing and interpreting clinical studies, as evidence suggests that early postmenopausal women produced different results than late menopausal women. One study found that dietary soy consumption affected gene expression differently than purified genistein [89] and provided strong proof about the difference in results after consumption of pure isoflavone versus soy flour, which may need to be considered during further studies.

*4.8. Genistein and miRNA*

In response to genistein administration, oncogenic miR-155 is repressed when cell viability reduces, whereas FOXO3, casein kinase, PTEN, and p27, the pro-apoptotic and anti-cell proliferative targets, are elevated [49,90]. As a result, miR-155 downregulation concomitantly aids in mammary cancer repression. Another micro-RNA, miR-23b, has been found to influence cytoskeletal rearrangement and contribute to PAK2-induced decreased invasion [50].

*4.9. Genistein and Estrogen*

Genistein, along with anti-estrogenic and anti-cancer properties, has also been noted to possess estrogen-like properties [91]. Given the structural similarity between genistein and estrogen, in circulation, it may exhibit a number of activities mimicking estrogen. It is known to act on both estrogen receptors α and β through the classical genomic mechanism [92]. However, it differs from estrogen in its preference for ER β.

So far, many meta-analyses which have been published have not been able to consistently conclude the nature of the relationship between genistein and breast cancer. While

some reports suggest the protective effect of soy consumption in premenopausal women compared to postmenopausal women, others have concluded no association between menopausal status, genistein, and breast cancer [93–95]. Yet other studies have suggested the protective effect of genistein, however, only in postmenopausal women [96]. Some studies have also suggested that due to difference in the levels of estrogen, the effects of menopausal status (i.e., premenopausal and postmenopausal women) play a modifying role in genistein—breast cancer association [97]. Furthermore, it has been suggested that genistein may be associated with increased survival rates in ER negative, ER+, and postmenopausal patients [98]. Some studies have found genistein-induced cell death in breast cancer cells irrespective of the presence or absence of estrogen [45,99]. A large study including breast cancer diagnosed Asian and American women found that consumption of soy every day significantly declined breast cancer reoccurrence as well as non-significantly reduced the risk [91]. Further conflicting evidence has been documented reporting that a subset of the population might be adversely affected through gene expression. Gene expression because of soy intake is characterized by an overexpression of FGFR2 and genes that drive cell cycle and proliferation pathways. However, the study period or the consumption period was for 1–4 weeks, which may be a drawback because patients might consume soy proteins for years [47,88].

Because genistein can only weakly bind to the estrogen receptor, it interfered with the binding inside estrogen molecules, causing ER-dependent pathways to be impacted in a dose-dependent manner [45,100,101]. In a dose-dependent manner, genistein could also stop the growth inhibition caused by aromatase inhibitor fadrozole [102]. ER $\alpha$ mRNA and protein expression in human breast cancer cells was found to be inhibited with sufficient doses [41]. Because estrogen is a primary promoter of breast cancer tumor growth, inhibiting it with genistein allows its consequences to be reduced, resulting in a reduction in tumor cell growth. Genistein has a greater affinity for Er$\beta$ than Er$\alpha$, providing a powerful feature of control of breast cancer development. Genistein enhanced c-fos expression both through ER $\alpha$ and through the G protein-coupled receptor homologue in an ER-independent way, as seen in ER $\alpha$-positive MCF7 and ER-negative SKBR3 breast cancer cells. c-fos proto-oncogene expression may be considered an early sensor of estrogenic activity in cells [103]. Further, study into the effect of genistein on the inflammation of cancerous cells with various different receptors $\alpha$ (ER$\alpha$) and $\beta$ (ER$\beta$) ratio revealed that genistein could modulate inflammatory-related genes though the help of ER [104].

Using transcriptomics and qualitative proteomics, the effects of ER$\alpha$ and ER$\beta$ on gene and protein expression in T47D cells treated with genistein were studied, revealing an interplay between focal adhesin kinase, actin, and integrins in signaling pathways in cells with lower levels of Er$\alpha$ and depleted levels of ER$\beta$. Further, in cells expressing Er$\alpha$, genistein was found to induce signatures of transcriptomics and proteomics which signaled rapid cell growth and migration. ER$\beta$ led to a decrease in motility of cells and cancer potential [105]. Other works have pointed towards the possibility that genistein modulates oxidative stress in cells according the ER$\alpha$ and Er$\beta$ ratios, causes cell cycle arrest, and leads to increased function of mitochondria and upregulation of uncoupling protein 2 and sirtuins [106,107].

### 4.10. Exposure to Genistein in Early Developmental Stages

Various studies have proven that exposure to genistein early in life may reduce the incidence of breast cancer [108]. Mammary terminal end buds are ducts found in young animals that include a large number of undifferentiated cells that are vulnerable to carcinogens. When young rats were given genistein, the number of terminal end buds dropped while the number of lobules increased [109,110]. Researchers determined that pre-pubertal and adult exposure to chemically produced breast cancer in genistein-protected rats must occur between birth and the pre-pubertal period of mammary gland development for genistein to be protective [111]. Researchers have concluded that genistein operates as a chemo-preventive drug during the pre-pubertal stage, which they believe corresponds

to the teenage period in human life [111]. Through these studies, the cellular mechanism of action of genistein has been observed to be through increased cell differentiation of the breast [111,112].

### 4.11. Clinical Trials

Despite the vast number of studies to understand the association of genistein with breast cancer, for the clinical application of genistein as a promising anti-cancer therapeutic agent, its mechanisms and targets need to be understood better. So far, genistein has been utilized in a number of human clinical trials for the treatment of cancer. Phase I and II clinical trials checking the efficacy of genistein combined with FOLFOX for treatment in colorectal cancer have documented a safe and tolerable use with notable results, warranting further clinical trials for use of genistein in combination with other drugs in cancer chemoprevention [113]. Similarly, phase II trials studying the efficacy of genistein in bladder cancer have noted a bimodal effect of genistein, being effective at lower doses and warranting further trials of genistein in synergy with other drugs [114]. Three clinical trials including the use of genistein in the treatment of breast cancer have been completed so far [115]. A phase I double-blinded trial evaluating the effect of soy isoflavone consumption for 84 days in healthy postmenopausal women concluded that its consumption was safe even at 900 mg per day [116]. More clinical trials examining the effects of genistein on women at early- and late-postmenstrual ages, as well as men, may be deemed necessary to gain more insight on the effects of genistein.

### 5. Conclusions

In summary, this article explores the various evidences of genistein being responsible for prevention, retardation, or blockage of breast cancer development. As per pre-clinical and clinical evidence, genistein exhibits clear dose-dependent anti-breast cancer effects achieved via a number of different molecular pathways, and based on these indications, it may be hypothesized that genistein could be a potent anti-breast cancer agent.

**Funding:** This research received no external funding.

**Institutional Review Board Statement:** Not applicable.

**Informed Consent Statement:** Not applicable.

**Acknowledgments:** RGA acknowledges the support and infrastructure provided by Department of Clinical Sciences, College of Veterinary Medicine, Kansas State University, Manhattan, KS, 66506, USA, SSB, CS, SKP acknowledge the infrastructure and support provided by the JSS Academy of Higher Education and Research (JSS AHER), Mysuru, India. KSP thankfully acknowledges the director Amrita Vishwa Vidyapeetham, Mysuru campus, for infrastructure support. AS thankfully acknowledges the the King Saud University, Saudi Arabia for support. PR thankfully acknowledges the Acharya Institute of technology for support, CAC thankfully acknowledges the Midwest Veterinary Services for support.

**Conflicts of Interest:** The authors declare no conflict of interest.

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
