# Peer review of "Genistein: A Potent Anti-Breast Cancer Agent"

_cimb, doi:10.3390/cimb43030106_

Round 1
Reviewer 1 Report
The review by Bhat et al., titled, “Genistein: A potent anti-breast cancer agent” fails to convince me as a reviewer on the significance and impact of the review. The review reads like a compilation of facts rather than a comprehensive analysis on the topic which will help advance knowledge in the field. I regret to conclude that the review may not be suitable for acceptance to publication. The review needs extensive revision. I suggest the authors rewrite and revise the manuscript. Overall multiple errors all throughout the text are detected in the manuscript and the manuscript needs a careful review.
Few examples are:
- In introduction: The authors quote, “It was first isolated from in 1899 and named after it”-Isolated from what?
- The authors quote, “Genistein possess a wide range of biomedical benefits which have been well documented including its ability to decrease various types of cancers”- Is Genistein prescribed for cancer treatment, or this conclusion is based on laboratory studies and Genistein have never advanced to clinic? Please provide evidence!
- The authors quote, “In humans, micromolar levels of genistein in blood can be found through prolonged dietary exposure (17)”. The reference 17 by Yang et al. is a study in mice and not humans!
- The authors quote, “Three clinical trials including the use of genistein in the treatment of breast cancer have been completed so far, but the findings have yet to be published (88)”- The reference list end in 87 and reference 88 does not exist.
Author Response
Response to reviewer #1:
- In introduction: The authors quote, “It was first isolated from in 1899 and named after it”-Isolated from what?
Response: We thank the reviewer for pointing out the typographic error. In the revised manuscript, the point has been amended in line number 24.
2. The authors quote, “Genistein possess a wide range of biomedical benefits which have been well documented including its ability to decrease various types of cancers”- Is Genistein prescribed for cancer treatment, or this conclusion is based on laboratory studies and Genistein have never advanced to clinic? Please provide evidence!
Response: Thanking the reviewer for the suggestion, we would like to inform that the statement is based on laboratory studies and relevant scientific evidences. Citations have been added, in line 28, to substantiate the same.
3. The authors quote, “In humans, micromolar levels of genistein in blood can be found through prolonged dietary exposure (17)”. The reference 17 by Yang et al. is a study in mice and not humans!
Response: We sincerely apologise for the inadvertent mistake and thank the reviewer for pointing this out. In the revised manuscript, the appropriate reference for the same has been updated in line 87.
4. The authors quote, “Three clinical trials including the use of genistein in the treatment of breast cancer have been completed so far, but the findings have yet to be published (88)”- The reference list end in 87 and reference 88 does not exist.
Response: Authors thank the reviewer for bringing this up. We have duly updated the bibliography as directed.

Reviewer 2 Report
cimb-1366879-Genistein: A potent anti-breast cancer agent
Brief summary
Bhat, Prasad and colleagues aimed to review the potential therapeutic value of the soy isoflavone ginestein in breast cancer. The authors presented a compilation of studies and described the main mechanisms by which ginestein may exert its therapeutic effects in breast cancer cells. However, the manuscript lacks citation to several other relevant studies on this topic, which could be included to refine the revision manuscript and differentiate it from others previously published.
Broad comments
- Figure 2 and tables 1 and 2 add significant value to the revision manuscript.
- The authors emphasize in the abstract the differential effects of genistein in women depending on the menopausal/menstrual ages but explore little the molecular similarity between genistein estrogen and the role of estrogen receptor on those differential effects. This discussion may be of particular importance in this revision article considering its focus in breast cancer and keeping in mind the utility of estrogen receptor expression to the classification of the diverse types of breast cancer. Some references regarding this point are thus missing and should be included in the discussion:
- Hwang C.S., Kwak H.S., Lim H.J., Lee S.H., Kang Y.S., Choe T.B., Hur H.G., Han K.O. Isoflavone metabolites and their in vitro dual functions: They can act as an estrogenic agonist or antagonist depending on the estrogen concentration. J. Steroid Biochem. 2006;101:246–253. doi: 10.1016/j.jsbmb.2006.06.020
- Sotoca A.M., Gelpke M.D., Boeren S., Strom A., Gustafsson J.A., Murk A.J., Rietjens I.M., Vervoort J. Quantitative proteomics and transcriptomics addressing the estrogen receptor subtype-mediated effects in T47D breast cancer cells exposed to the phytoestrogen genistein. Mol. Cell. Proteomics. 2011;10:M110.002170. doi: 10.1074/mcp.M110.002170
- van Duursen et al, Toxicology. 2011 Nov 18;289(2-3):67-73. doi: 10.1016/j.tox.2011.07.005. Genistein induces breast cancer-associated aromatase and stimulates estrogen-dependent tumor cell growth in in vitro breast cancer model
- Nadal-Serrano M., Pons D.G., Sastre-Serra J., Blanquer-Rosselló M.d.M., Roca P., Oliver J. Genistein modulates oxidative stress in breast cancer cell lines according to ERα/ERβ ratio: Effects on mitochondrial functionality, sirtuins, uncoupling protein 2 and antioxidant enzymes. Int. J. Biochem. Cell B. 2013;45:2045–2051. doi: 10.1016/j.biocel.2013.07.002
- Pons D.G., Nadal-Serrano M., Blanquer-Rossello M.M., Sastre-Serra J., Oliver J., Roca P. Genistein modulates proliferation and mitochondrial functionality in breast cancer cells depending on ERalpha/ERbeta ratio. J. Cell. Biochem. 2014;115:949–958. doi: 10.1002/jcb.24737
- Introduction: for several times the authors refer to previous studies, and because of that they could include some references, at least to some of the review articles already published as they mention.
- Section 4b would be more comprehensive for the reader if the authors sub-divide it into dedicated sections according for instance with the mechanism of action of genistein cellular effects (e.g. oxidative stress, cell proliferation and survival, apoptosis, angiogenesis, metastasis, on specific signaling pathways)
- Section 4c. Authors refer that “More clinical trials examining the effects of genistein on women and men at early postmenstrual and late postmenstrual ages may provide more insight on the effects of genistein.” However, this should be corrected by at least switching the words women and men, since in principle, men do not live “early postmenstrual and late postmenstrual ages”.
- Conclusion: authors discuss that “…metabolomic studies may be required in order to assess the intracellular concentrations of genistein…” but should also cite at least the references:
- Uifălean A., Schneider S., Gierok P., Ionescu C., Iuga C., Lalk M. The impact of soy isoflavones on MCF-7 and MDA-MB-231 breast cancer cells using a global metabolomic approach. International Journal of Molecular Sciences. 2016;17(9):p. 1443. doi: 10.3390/ijms17091443.
- Poschner et al, Front Pharmacol. 2017 Oct 5;8:699. doi: 10.3389/fphar.2017.00699. eCollection 2017 The Impacts of Genistein and Daidzein on Estrogen Conjugations in Human Breast Cancer Cells: A Targeted Metabolomics Approach
- Conclusion: I not understand the criteria for the authors to suddenly suggest Nrf2 pathway in the sentence “The role of genistein in the Nrf2 pathway can be investigated further, which can provide a better insight into pathways in cancer.”
- Some studies suggest a role of genistein in breast cancer stem cells but the authors only briefly mention it. Some references that could also be included in this regard:
- Montales MT, Rahal OM, Nakatani H, Matsuda T, Simmen RC. Repression of mammary adipogenesis by genistein limits mammosphere formation of human MCF-7 cells. J Endocrinol. 2013;218:135–149.
- Montales MT, Rahal OM, Kang J, Rogers TJ, Prior RL, Wu X, et al. Repression of mammosphere formation of human breast cancer cells by soy isoflavone genistein and blueberry polyphenolic acids suggests diet-mediated targeting of cancer stem-like/progenitor cells. Carcinogenesis. 2012;33:652–660.
- Some studies also suggest that cytochrome P450 is involved in genistein effects on breast cancer cells but the authors do not refer to it in the manuscript.
Specific comments
- The authors should include a graphical abstract as proposed in the author guidelines.
- In vivo, in vitro and in silico should be italicized.
- References 6 and 7 are the same.
- A list of abbreviations should be included as a footnote in tables 1 and 2.
- Several other relevant references are missing in this manuscript (list not exhaustive):
- Wei Y. K., Gamra I., Davenport A., Lester R., Zhao L., Wei Y. Genistein induces cytochrome P450 1B1 gene expression and cell proliferation in human breast cancer MCF-7 cells. Journal of Environmental Pathology, Toxicology and Oncology : Official Organ of the International Society for Environmental Toxicology and Cancer. 2015;34(2):153–159. doi: 10.1615/JEnvironPatholToxicolOncol.2015013315.
- Yang X., Belosay A., Hartman J. A., et al. Dietary soy isoflavones increase metastasis to lungs in an experimental model of breast cancer with bone micro-tumors. Clinical & Experimental Metastasis. 2015;32(4):323–333. doi: 10.1007/s10585-015-9709-2
- Liu Y., Hilakivi-Clarke L., Zhang Y., et al. Isoflavones in soy flour diet have different effects on whole-genome expression patterns than purified isoflavone mix in human MCF-7 breast tumors in ovariectomized athymic nude mice. Molecular Nutrition & Food Research. 2015;59(8):1419–1430. doi: 10.1002/mnfr.201500028
- Uifălean A., Schneider S., Ionescu C., Lalk M., Iuga C. Soy isoflavones and breast cancer cell lines: molecular mechanisms and future perspectives. Molecules (Basel, Switzerland) 2016;21(1):p. 13. doi: 10.3390/molecules21010013
- Individual factors define the overall effects of dietary genistein exposure on breast cancer patients. Liu R, Yu X, Chen X, Zhong H, Liang C, Xu X, Xu W, Cheng Y, Wang W, Yu L, Wu Y, Yan N, Hu X. Nutr Res. 2019 Jul;67:1-16. doi: 10.1016/j.nutres.2019.03.015. Epub 2019 Mar 29
- Genistein: An Integrative Overview of Its Mode of Action, Pharmacological Properties, and Health Benefits. Sharifi-Rad J, Quispe C, Imran M, Rauf A, Nadeem M, Gondal TA, Ahmad B, Atif M, Mubarak MS, Sytar O, Zhilina OM, Garsiya ER, Smeriglio A, Trombetta D, Pons DG, Martorell M, Cardoso SM, Razis AFA, Sunusi U, Kamal RM, Rotariu LS, Butnariu M, Docea AO, Calina D. Oxid Med Cell Longev. 2021 Jul 19;2021:3268136. doi: 10.1155/2021/3268136. eCollection 2021
Author Response
Response to reviewer #2:
- Figure 2 and tables 1 and 2 add significant value to the revision manuscript.
Response: We thank the reviewer for suggestions, the figure and tables have been amended with the necessary information, as prescribed.
2. The authors emphasize in the abstract the differential effects of genistein in women depending on the menopausal/menstrual ages but explore little the molecular similarity between genistein estrogen and the role of estrogen receptor on those differential effects. This discussion may be of particular importance in this revision article considering its focus in breast cancer and keeping in mind the utility of estrogen receptor expression to the classification of the diverse types of breast cancer. Some references regarding this point are thus missing and should be included in the discussion:
Response: We are pleased to inform the reviewer that the necessary citations have been incorporated (as highlighted in yellow). The same can be found below for kind perusal.
- Hwang C.S., Kwak H.S., Lim H.J., Lee S.H., Kang Y.S., Choe T.B., Hur H.G., Han K.O. Isoflavone metabolites and their in vitro dual functions: They can act as an estrogenic agonist or antagonist depending on the estrogen concentration. J. Steroid Biochem. 2006;101:246–253. doi: 10.1016/j.jsbmb.2006.06.020
- Sotoca A.M., Gelpke M.D., Boeren S., Strom A., Gustafsson J.A., Murk A.J., Rietjens I.M., Vervoort J. Quantitative proteomics and transcriptomics addressing the estrogen receptor subtype-mediated effects in T47D breast cancer cells exposed to the phytoestrogen genistein. Mol. Cell. Proteomics. 2011;10:M110.002170. doi: 10.1074/mcp.M110.002170
- van Duursen et al, Toxicology. 2011 Nov 18;289(2-3):67-73. doi: 10.1016/j.tox.2011.07.005. Genistein induces breast cancer-associated aromatase and stimulates estrogen-dependent tumor cell growth in in vitro breast cancer model
- Nadal-Serrano M., Pons D.G., Sastre-Serra J., Blanquer-Rosselló M.d.M., Roca P., Oliver J. Genistein modulates oxidative stress in breast cancer cell lines according to ERα/ERβ ratio: Effects on mitochondrial functionality, sirtuins, uncoupling protein 2 and antioxidant enzymes. Int. J. Biochem. Cell B. 2013;45:2045–2051. doi: 10.1016/j.biocel.2013.07.002
- Pons D.G., Nadal-Serrano M., Blanquer-Rossello M.M., Sastre-Serra J., Oliver J., Roca P. Genistein modulates proliferation and mitochondrial functionality in breast cancer cells depending on ERalpha/ERbeta ratio. J. Cell. Biochem. 2014;115:949–958. doi: 10.1002/jcb.24737
3. Introduction: for several times the authors refer to previous studies, and because of that they could include some references, at least to some of the review articles already published as they mention.
Response: Thank you for this useful suggestion, the appropriate review article references have been updated as highlighted with the colour yellow in line 28.
4. Section 4b would be more comprehensive for the reader if the authors sub-divide it into dedicated sections according for instance with the mechanism of action of genistein cellular effects (e.g. oxidative stress, cell proliferation and survival, apoptosis, angiogenesis, metastasis, on specific signaling pathways)
Response: We thank the reviewer for prompt insight. As suggested, the section 4b has been modified in to subsections, based on the mechanism of action of genistein.
5. Section 4c. Authors refer that “More clinical trials examining the effects of genistein on women and men at early postmenstrual and late postmenstrual ages may provide more insight on the effects of genistein.” However, this should be corrected by at least switching the words women and men, since in principle, men do not live “early postmenstrual and late postmenstrual ages”.
Response: Thank you for bringing this to our attention. In the revised manuscript, the section 4c has been modified to say “More clinical trials examining the effects of genistein on women, at early- and late-postmenstrual ages, as well as men”.
6. Conclusion: authors discuss that “…metabolomic studies may be required in order to assess the intracellular concentrations of genistein…” but should also cite at least the references:
- Uifălean A., Schneider S., Gierok P., Ionescu C., Iuga C., Lalk M. The impact of soy isoflavones on MCF-7 and MDA-MB-231 breast cancer cells using a global metabolomic approach. International Journal of Molecular Sciences. 2016;17(9):p. 1443. doi: 10.3390/ijms17091443.
- Poschner et al, Front Pharmacol. 2017 Oct 5;8:699. doi: 10.3389/fphar.2017.00699. eCollection 2017 The Impacts of Genistein and Daidzein on Estrogen Conjugations in Human Breast Cancer Cells: A Targeted Metabolomics Approach
Response: We are pleased to inform that the necessary references suggested by the reviewer for the metabolomic studies have been added.
7. Conclusion: I not understand the criteria for the authors to suddenly suggest Nrf2 pathway in the sentence “The role of genistein in the Nrf2 pathway can be investigated further, which can provide a better insight into pathways in cancer.”
Response: We thank the reviewer for kind suggestion. We have incorporated the necessary changes as directed.
8. Some studies suggest a role of genistein in breast cancer stem cells but the authors only briefly mention it. Some references that could also be included in this regard:
- Montales MT, Rahal OM, Nakatani H, Matsuda T, Simmen RC. Repression of mammary adipogenesis by genistein limits mammosphere formation of human MCF-7 cells. J Endocrinol. 2013;218:135–149.
- Montales MT, Rahal OM, Kang J, Rogers TJ, Prior RL, Wu X, et al. Repression of mammosphere formation of human breast cancer cells by soy isoflavone genistein and blueberry polyphenolic acids suggests diet-mediated targeting of cancer stem-like/progenitor cells. Carcinogenesis. 2012;33:652–660.
Response: Thanking the reviewer, we are happy to inform that the effect of genistein on breast cancer stem cells has now been discussed more detail.
9. Some studies also suggest that cytochrome P450 is involved in genistein effects on breast cancer cells but the authors do not refer to it in the manuscript.
Response: We have gladly incorporated the suggested addition.
Specific comments
10. The authors should include a graphical abstract as proposed in the author guidelines.
Response: Thanking the reviewers, we would like to mention that a graphical abstract has been added.
11. In vivo, in vitro and in silico should be italicized.
Response: We have done the needful as suggested.
12. References 6 and 7 are the same.
Response: Thank you, the bibliography has been modified.
- A list of abbreviations should be included as a footnote in tables 1 and 2.
Response: Thanking the reviewer for the valuable suggestion, we have made sure to include a list of abbreviations as footnote for each table.
13. Several other relevant references are missing in this manuscript (list not exhaustive):
- Wei Y. K., Gamra I., Davenport A., Lester R., Zhao L., Wei Y. Genistein induces cytochrome P450 1B1 gene expression and cell proliferation in human breast cancer MCF-7 cells. Journal of Environmental Pathology, Toxicology and Oncology: Official Organ of the International Society for Environmental Toxicology and Cancer. 2015;34(2):153–159. doi: 10.1615/JEnvironPatholToxicolOncol.2015013315.
- Yang X., Belosay A., Hartman J. A., et al. Dietary soy isoflavones increase metastasis to lungs in an experimental model of breast cancer with bone micro-tumors. Clinical & Experimental Metastasis. 2015;32(4):323–333. doi: 10.1007/s10585-015-9709-2
- Liu Y., Hilakivi-Clarke L., Zhang Y., et al. Isoflavones in soy flour diet have different effects on whole-genome expression patterns than purified isoflavone mix in human MCF-7 breast tumors in ovariectomized athymic nude mice. Molecular Nutrition & Food Research. 2015;59(8):1419–1430. doi: 10.1002/mnfr.201500028
- Uifălean A., Schneider S., Ionescu C., Lalk M., Iuga C. Soy isoflavones and breast cancer cell lines: molecular mechanisms and future perspectives. Molecules (Basel, Switzerland) 2016;21(1):p. 13. doi: 10.3390/molecules21010013
- Individual factors define the overall effects of dietary genistein exposure on breast cancer patients. Liu R, Yu X, Chen X, Zhong H, Liang C, Xu X, Xu W, Cheng Y, Wang W, Yu L, Wu Y, Yan N, Hu X. Nutr Res. 2019 Jul;67:1-16. doi: 10.1016/j.nutres.2019.03.015. Epub 2019 Mar 29
- Genistein: An Integrative Overview of Its Mode of Action, Pharmacological Properties, and Health Benefits. Sharifi-Rad J, Quispe C, Imran M, Rauf A, Nadeem M, Gondal TA, Ahmad B, Atif M, Mubarak MS, Sytar O, Zhilina OM, Garsiya ER, Smeriglio A, Trombetta D, Pons DG, Martorell M, Cardoso SM, Razis AFA, Sunusi U, Kamal RM, Rotariu LS, Butnariu M, Docea AO, Calina D. Oxid Med Cell Longev. 2021 Jul 19;2021:3268136. doi: 10.1155/2021/3268136. eCollection 2021
Response: Thanking the reviewer, we duly accept the reviewer’s keen interest in strengthening our article. The suggested references have been incorporated as highlighted in the colour yellow.

Round 2
Reviewer 1 Report
Please add a section describing clearly that the review is based on in vitro and in vivo studies on cell lines and animals and no actual evidence of the role of Genistein as an anti-cancer agent in humans in the clinic. The tile ," Genistein: A potent anti-breast cancer agent" is misleading and should clearly distinguish that the scientific observations and conclusions are based on pre-clinical studies.
Author Response
Response to reviewer 1:
Please add a section describing clearly that the review is based on in vitro and in vivo studies on cell lines and animals and no actual evidence of the role of Genistein as an anti-cancer agent in humans in the clinic. The title," Genistein: A potent anti-breast cancer agent" is misleading and should clearly distinguish that the scientific observations and conclusions are based on pre-clinical studies.
Response: Thanking the reviewer for the suggestion, we would like to inform that in line number 34 (as highlighted in yellow), it has been now mentioned that the review is based on the available in vitro and in vivo evidences.
However, in the clinical trials section (Line 428-444), evidence from clinical trials of the efficacy of genistein in the chemoprevention of different cancers at various phases have been added. As per pre-clinical and clinical evidences, it may not be misleading to say that genistein could be a potent anti-breast cancer agent.